# The Confirmation of a Ploidy Periclinal Chimera of the Meiwa Kumquat (*Fortunella crassifolia* Swingle) Induced by Colchicine Treatment to Nucellar Embryos and Its Morphological Characteristics

**Tsunaki Nukaya [1,†], Miki Sudo [1,†], Masaki Yahata [1,\*], Tomohiro Ohta [1], Akiyoshi Tominaga [1], Hiroo Mukai [1], Kiichi Yasuda [2] and Hisato Kunitake [3]**

[1] Faculty of Agriculture, Shizuoka University, Shizuoka 422-8529, Japan; t_nukaya54@yahoo.co.jp (T.N.); sudo.miki@shizuoka.ac.jp (M.S.); tomohiro1_ota@pref.shizuoka.lg.jp (T.O.); tominaga.akiyoshi@shizuoka.ac.jp (A.T.); mukai.hiroo@shizuoka.ac.jp (H.M.)
[2] School of Agriculture, Tokai University, Kumamoto 862-8652, Japan; yk964422@tsc.u-tokai.ac.jp
[3] Faculty of Agriculture, University of Miyazaki, Miyazaki 889-2192, Japan; hkuni@cc.miyazaki-u.ac.jp
\* Correspondence: yahata.masaki@shizuoka.ac.jp; Tel.: +81-54-641-9500
† These authors contributed equally to the article.

**Abstract:** A ploidy chimera of the Meiwa kumquat (*Fortunella crassifolia* Swingle), which had been induced by treating the nucellar embryos with colchicine, and had diploid (2n = 2x = 18) and tetraploid (2n = 4x = 36) cells, was examined for its ploidy level, morphological characteristics, and sizes of its cells in its leaves, flowers, and fruits to reveal the ploidy level of each histogenic layer. Furthermore, the chimera was crossed with the diploid kumquat to evaluate the ploidy level of its reproductive organs. The morphological characteristics and the sizes of the cells in the leaves, flowers, and fruits of the chimera were similar to those of the tetraploid Meiwa kumquat and the ploidy periclinal chimera known as "Yubeni," with diploids in the histogenic layer I (L1) and tetraploids in the histogenic layer II (L2) and III (L3). However, the epidermis derived from the L1 of the chimera showed the same result as the diploid Meiwa kumquat in all organs and cells. The sexual organs derived from the L2 of the chimera were significantly larger than those of the diploid. Moreover, the ploidy level of the seedlings obtained from the chimera was mostly tetraploid. In the midrib derived from the L3, the chimera displayed the fluorescence intensity of a tetraploid by flow cytometric analysis and had the same size of the cells as the tetraploid and the Yubeni. According to these results, the chimera is thought to be a ploidy periclinal chimera with diploid cells in the outermost layer (L1) and tetraploid cells in the inner layers (L2 and L3) of the shoot apical meristem. The chimera had desirable fruit traits for a kumquat such as a thick pericarp, a high sugar content, and a small number of developed seeds. Furthermore, triploid progenies were obtained from reciprocal crosses between the chimera and diploid kumquat.

**Keywords:** anatomy; citrus; flow cytometry; histogenic layer; polyploidy breeding

## 1. Introduction

The shoot apical meristem of higher plants consists of three histogenic layers [1]. This is known as the "Tunica-Corpus" theory; i.e., the shoot apical meristem consists of L1 (Tunica) and L2/L3 (Corpus). In the genus *Citrus* and its related genera, the histogenic layers are differentiated by their parts: the dermal system (guard cell and juice sac) in L1, parenchyma and reproductive organs (mesophyll cell, pollen and seed) in L2, and the vascular bundle (cambium and pith) in L3, respectively [2]. A plant

is considered to be a chimera if it has two or more genetic constitutions in its shoot apical meristem, and chimeras are classified into three types: sectorial, periclinal, and mericlinal [3,4]. Sectorial chimeras have a sector of all cell layers that is genetically different. Periclinal chimeras are chimeras in which one or more entire cell layer(s) is genetically distinct from another cell layer. Mericlinal chimeras have part of one or more layers that is genetically different.

In the genus *Citrus*, spontaneously arising chimeras have been reported previously; i.e., autogenous chimeras such as the "Suzuki wase" satsuma mandarin (*C. unshiu* Marcow.) and the "Thompson" grapefruit (*C. paradisi* Macfad.) [5,6] and graft chimeras such as the "Kobayashi mikan" [Natsudaidai (*C. natsudaidai* Hayata) + Satsuma mandarin], the "Kinkoji unshiu" [Kinkoji (*C. obovoidea* hort. ex Takahashi) + Satsuma mandarin], and the "Zaohong" navel orange ["Robertson" navel orange (*C. sinensis* Osbeck var. *Brasiliensis* Tanaka) + Satsuma mandarin] [7,8] are well known. All of these cultivars are periclinal chimeras. In recent years, production of synthetic graft chimeras has been successful [9,10], and many periclinal graft chimera cultivars have been registered in Japan. Furthermore, 2x+4x ploidy chimeras were produced by treating the apical meristems, undeveloped ovules, calluses, and protoplasts with antimitotic agents [11]. These were then used as breeding materials for triploid production in the genus *Citrus*.

In kumquats (*Fortunella* spp.), Yubeni which have a large fruit size and a high sugar content was discovered to be a bud mutation of the Meiwa kumquat. Yasuda et al. [12] demonstrated that Yubeni was a ploidy periclinal chimera with diploid cells in the outermost layer (L1) and tetraploid cells in the inner layers (L2 and L3) of the shoot apical meristem, and showed that the Yubeni increase in fruit quality was due to the tetraploidization in L2 and L3. Furthermore, this result showed that the 2x–4x–4x ploidy periclinal chimera kumquat could be used both for the parental line in triploid breeding and for direct domestication. However, since the ploidy chimera kumquat has rarely been reported [12], information on their characteristics is lacking. To prove the future usefulness of the ploidy chimera kumquat, it is necessary to evaluate more kinds and collect information.

Yahata et al. [13] produced many tetraploid Meiwa kumquats by applying a colchicine treatment to nucellar embryos. After they were grafted onto the trifoliate oranges [*Poncirus trifoliata* (L.) Raf.], they showed vigorous growth and they flowered and fruited for the first time, five years after budding. Nukaya et al. [14] examined whether these plants maintained tetraploidy, and found a 2x+4x ploidy chimera among their tetraploids. If this ploidy chimera is a ploidy peripheral chimera with the same histogenic layer as the Yubeni (layer constitution: L1–L2–L3 = 2x–4x–4x), it would be very useful in future kumquat breeding.

To clarify the ploidy level of each histogenic layer in the 2x+4x ploidy chimera, in the present study, we performed ploidy analysis by flow cytometry, cell observation using histological techniques and morphological characteristics in several tissues and organs of this ploidy chimera, and an evaluation of the reproductive organs of the chimera by crossing it with the diploid kumquat.

## 2. Materials and Methods

### 2.1. Plant Materials

The 2x+4x ploidy chimera, induced by applying a colchicine treatment to nucellar embryos of the Meiwa kumquat [14], was used in the present study. The original diploid Meiwa kumquat, the tetraploid induced by treating nucellar embryos of the Meiwa kumquat with colchicine, and the ploidy periclinal chimera mutant Yubeni, which originated from the Meiwa kumquat and has diploids in L1 and tetraploids in L2 and L3, were used as the control. These plant materials were grafted onto the trifoliate orange and grown in 45 L containers for approximately five years in the greenhouse of the Faculty of Agriculture, Shizuoka University, Shizuoka, Japan.

### 2.2. Confirmation of Ploidy Level by Flow Cyotometry

Approximately 50 mg segments of whole leaf, midrib, petal, filament, style, ovary, seed, juice sac, albedo, and flavedo were collected from each plant material. Their samples were chopped with a razor blade and blended for 5 min with a 2 mL buffer solution containing 1.0% (*v/v*) Triton X-100, 140 mM mercaptoethanol, 50 mM $Na_2SO_3$, and 50 mM Tris-HCl at pH 7.5, according to the preparation method of Yahata et al. [15]. An aliquot (550 μL) of each sample was filtered through Miracloth (Merck KGaA, Drarmstadt, Germany), and the filtrate was stained with 50 μL of 0.5 g $L^{-1}$ propidium iodide (PI). The relative fluorescence intensity of the nuclear DNA was measured with a flow cytometry system (FCM, EPICS XL; Beckman Coulter, Inc., Pasadena, CA, USA) equipped with an argon laser (488 nm, 15 mW).

### 2.3. The Characteristics of Leaves, Flowers, Pollen, and Fruits in the Chimera

The characteristics of fully expanded leaves (e.g., blade size, thickness, guard cell size, guard cell density, and cell sizes) and flowers just before bloom (e.g., the size of the flower bud, petal, pistil, ovary, and pollen, and the number of petals and stamens) were measured. Guard cells and pollen grains were observed using a scanning electron microscope (Miniscope® TM3030Plus, Hitachi High-Technologies, Tokyo, Japan). The epidermises, palisade parenchymas, spongy parenchymas, vessels, and sieve tubes were used to measure the cell size. According to the methods of Nii et al. [16], their semi-ultra thin sections (1.5 μm) were cut using a glass knife before staining with methylene blue for histological examination under an optical microscope BX51 (Olympus Co., Ltd., Tokyo, Japan). Images were taken with a DP70 digital camera (Olympus). The weight, size, pericarp weight, the number of locules and seeds, soluble solids content (SSC), and titratable acidity (TA) of each kind of fruit were measured at the end of January. Each measurement used 20 samples.

Pollen fertility was evaluated by stainability and in vitro germination. Pollen stainability was estimated by staining the samples with 1% acetocarmine after squashing nearly mature anthers on a glass slide. In vitro germination of the pollen grains was performed on microscope slides covered with a 2 mm layer of 1% (*w/v*) agar medium containing 10% sucrose. Five stamens, each from different flowers, were rubbed on the agar medium, and the slides were then incubated for 10 h in a moistened chamber at 25 °C in the dark. Each test was evaluated from 500 grains with five repetitions.

### 2.4. Crossing for the Evaluation of the Reproductive Organs of the Chimera

The cross combinations are shown in Table 6. The flowers were pollinated immediately after emasculation and covered with paraffin paper bags. Seeds were collected from each mature fruit of all the crosses and were classified according to their size and shape into two groups: developed or undeveloped. After being numbered and weighed, both the developed and undeveloped seeds were cultured on Murashige and Skoog (MS) medium [17] containing 500 mg $L^{-1}$ malt extract, 30 g $L^{-1}$ sucrose, and 2 g $L^{-1}$ gellan gum at 25 °C under continuous illumination (38 μmol $m^{-2}$ $s^{-1}$). After germination, the seedlings were transplanted into vermiculite in pots and were transferred to a greenhouse. Ploidy analysis of the seedlings was performed by FCM using young leaves and chromosome observation using root tips according to the methods of Fukui [18] with some modifications.

## 3. Results

### 3.1. Confirmation of the Ploidy Level by FCM

The fluorescence intensities of all the examined tissues and organs of the diploid and the tetraploid Meiwa kumquats showed only diploid and tetraploid DNA values, respectively. In the 2x+4x ploidy chimera (Table 1, Figure 1), the whole leaf, petal, filament, style, ovary, and flavedo had both a diploid peak and a tetraploid peak, although the juice sac was diploid, and the midrib, seed, and albedo were tetraploid, respectively. Furthermore, the chimera showed the same fluorescence intensities as the Yubeni.

**Table 1.** Flow cytometric analysis of each organ and tissue in the diploid, the tetraploid, the Yubeni, and the chimera of the Meiwa kumquat.

|  | Leaf | | Flower | | | | | Fruit | | |
|---|---|---|---|---|---|---|---|---|---|---|
|  | Whole | Midrib | Petal | Filament | Style | Ovary | Seed | Juice Sac | Albedo | Flavedo |
| Diploid | 2x | 2x | 2x | 2x | 2x | 2x | 2x | 2x | 2x | 2x |
| Tetraploid | 4x | 4x | 4x | 4x | 4x | 4x | 4x | 4x | 4x | 4x |
| Yubeni | 2x+4x | 4x | 2x+4x | 2x+4x | 2x+4x | 2x+4x | 4x | 2x | 4x | 2x+4x |
| Chimera | 2x+4x | 4x | 2x+4x | 2x+4x | 2x+4x | 2x+4x | 4x | 2x | 4x | 2x+4x |

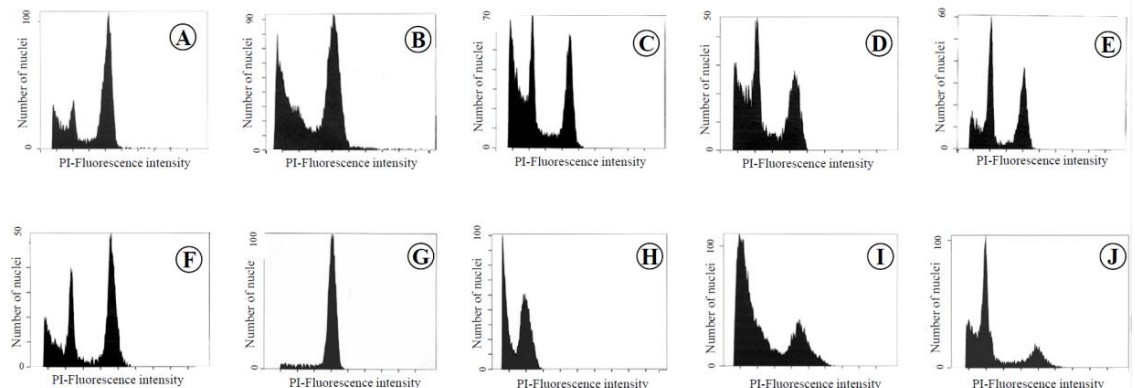

**Figure 1.** Flow cytometric analysis of each organ and tissue in leaves, flowers, seeds, and fruits of the chimera of the Meiwa kumquat. **A**: Leaf, **B**: Midrib, **C**: Petal, **D**: Filament, **E**: Style, **F**: Ovary, **G**: Seed, **H**: Juice sac, **I**: Albedo, **J**: Flavedo.

*3.2. The Characteristics of Leaves, Flowers, Pollen, and Fruits in the Chimera*

The morphological characteristics of the chimera were compared with those of the diploid, the tetraploid, and the Yubeni. The chimera had significantly rounder and thicker leaves as compared to those of the diploid, and it was mostly equal to those of the tetraploid and the Yubeni (Table 2, Figure 2A, Figure 3). On the other hand, the size and density of guard cells in the chimera was mostly equal to those of the diploid and the Yubeni (Figure 4). The epidermic cell size of the chimera showed the same value as the diploid and the Yubeni (Table 3, Figure 3). On the other hand, the palisade parenchyma and spongy parenchyma cells of the chimera were shown to be mostly the same sizes as the tetraploid and the Yubeni. Additionally, there were no differences in the cell sizes of the vessels and sieve tubes in midribs among the tetraploid, the Yubeni, and the chimera (Figure 5).

**Table 2.** Comparison of morphological characteristics of leaves in the diploid, the tetraploid, the Yubeni, and the chimera of the Meiwa kumquat.

|  | Ploidy Level | Leaf Blade (mm) | | Shape Index of Leaf [y] | Leaf Thickness (μm) | Guard Cell (μm) | | Guard Cell Density (No./mm²) |
|---|---|---|---|---|---|---|---|---|
|  |  | Length | Width |  |  | Length | Width |  |
| Diploid | 2x | 75.2 a [z] | 27.7 c | 2.7 a | 437.0 b | 22.0 c | 19.6 b | 397.4 a |
| Tetraploid | 4x | 70.0 b | 32.3 b | 2.2 b | 519.6 a | 27.6 a | 23.9 a | 289.5 b |
| Yubeni | 2x+4x | 72 a b | 32.9 b | 2.2 b | 532.3 a | 24.2 b | 19.3 b | 419.3 a |
| Chimera | 2x+4x | 75.8 a | 39.6 a | 1.9 c | 523.7 a | 23.2 c | 19.8 b | 375.8 a |

[y] Length of leaf blade/width of leaf blade; [z] different letters represent significant differences in Tukey's multiple test, 1% level.

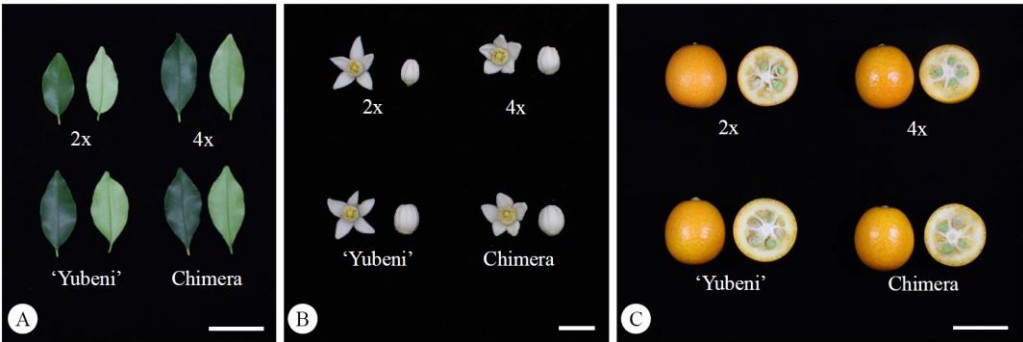

**Figure 2.** Comparison of the morphological characteristics of leaves (**A**, bar = 5 cm), flowers (**B**, bar = 1 cm), and fruits (**C**, bar = 3 cm) in the diploid, the tetraploid, the Yubeni, and the chimera of the Meiwa kumquat.

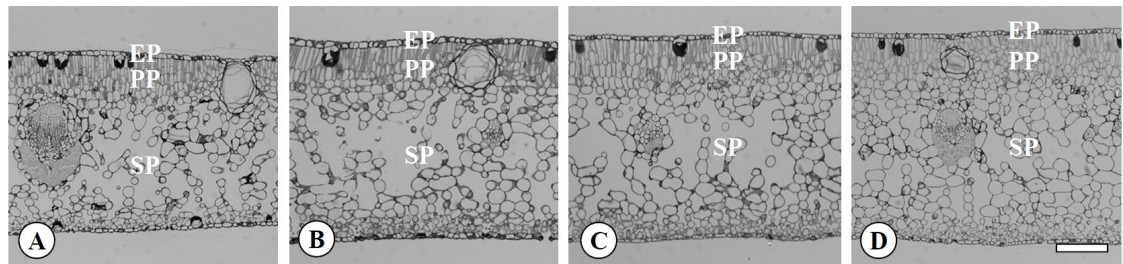

**Figure 3.** Transversal sections of leaves in the diploid (**A**), the tetraploid (**B**), the Yubeni (**C**), and the chimera (**D**) of the Meiwa kumquat. Bar = 100 μm. EP: Epidermis, PP: Palisade parenchyma, SP: Spongy parenchyma.

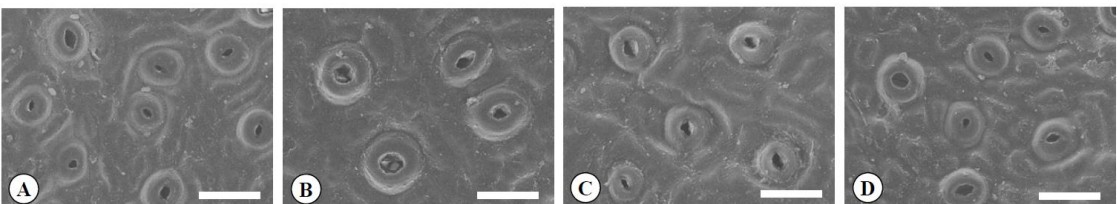

**Figure 4.** Scanning electron micrographs of guard cells in the diploid (**A**), the tetraploid (**B**), the Yubeni (**C**), and the chimera (**D**) of the Meiwa kumquat. Bars = 30 μm.

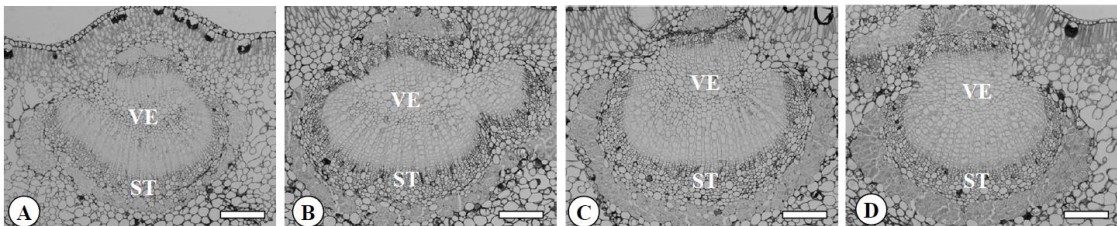

**Figure 5.** Transversal sections of midribs in the diploid (**A**), the tetraploid (**B**), the Yubeni (**C**), and the chimera (**D**) of the Meiwa kumquat. Bars = 100 μm. VE: Vessel, ST: Sieve tube.

**Table 3.** Comparison of the cell sizes of leaves in the diploid, the tetraploid, the Yubeni, and the chimera of the Meiwa kumquat.

| | Ploidy Level | Epidermis (μm) | | Palisade Parenchyma (μm) | | Spongy Parenchyma (μm) | | Vessel (μm) | | Sieve Tube (μm) | |
|---|---|---|---|---|---|---|---|---|---|---|---|
| | | Major Axis | Minor Axis | Major Axis | Minor Axis | Major Axis | Minor Axis | Major Axis | Minor Axis | Major Axis | Minor Axis |
| Diploid | 2x | 19.6 b [z] | 13.5 b | 27.5 b | 8.9 b | 35.5 b | 25.5 b | 18.6 b | 16.9 b | 19.0 b | 15.1 b |
| Tetraploid | 4x | 27.4 a | 15.9 a | 41.7 a | 13.0 a | 60.4 a | 40.2 a | 26.2 a | 22.6 a | 22.0 a | 18.0 a |
| Yubeni | 2x+4x | 16.0 c | 11.9 c | 42.6 a | 11.5 a | 54.2 a | 34.9 a | 26.3 a | 22.0 a | 24.4 a | 18.8 a |
| Chimera | 2x+4x | 16.8 c | 11.4 c | 46.0 a | 11.6 a | 54.2 a | 35.0 a | 25.8 a | 22.5 a | 22.2 a | 18.3 a |

[z] Different letters represent significant differences in Tukey's multiple test, 1% level.

The chimera had significantly larger flower buds and ovaries as compared to those of the diploid (Table 4, Figure 2B). No difference in flower morphology was observed among the tetraploid, the Yubeni, and the chimera. The average size of the pollen grains from the chimera was larger than that of the grains from the diploid (Table 4, Figure 6). The pollen fertility of the chimera was significantly lower than that of the diploid and was about the same as those of the tetraploid and the Yubeni.

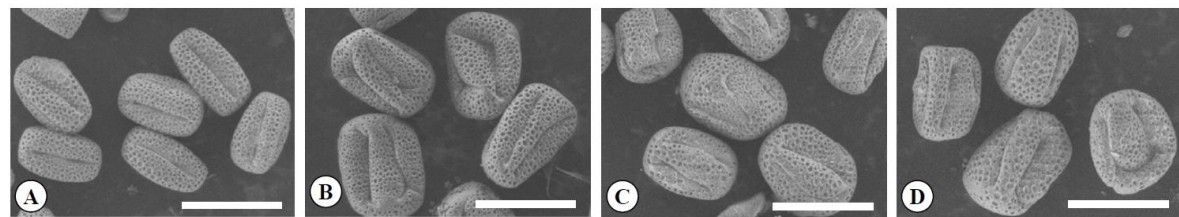

**Figure 6.** Scanning electron micrographs of pollen grains in the diploid (**A**), the tetraploid (**B**), the Yubeni (**C**), and the chimera (**D**) of the Meiwa kumquat. Bars = 30 μm.

There was no significant difference in the size of the fruits (Table 5, Figure 2C). However, the percentage of the pericarp weight per fruit in the tetraploid, the Yubeni, and the chimera was significantly higher in comparison to that of the diploid. The average number of seeds per fruit obtained from the diploid was 5.7, whereas that of the chimera was significantly lesser at 2.6. SSC in the pericarp and juice sac of the chimera was significantly higher than that of the diploid.

*3.3. Crossing for the Evaluation of the Reproductive Organs of the Chimera*

In order to evaluate the ploidy level in the reproductive organs of the chimera, crossing with the diploid Meiwa kumquat was carried out (Table 6). When the diploid was pollinated with pollen of the chimera, the frequency of developed seeds (23.1%) was lower than that of the self-pollinated fruit (86.8%). On the other hand, the frequency of developed seeds in the chimera was 86.7% when reverse-crossed, whereas that in the self-pollinated fruit was 89.2%. These developed seeds cultured on an MS medium germinated normally. The ploidy levels of these seedlings were confirmed by FCM analysis and chromosome observation (Table 6, Figure 7). Consequently, 8 and 12 triploid seedlings were obtained from crosses between the diploid and the chimera and from the reverse cross, respectively (Figure 8). Moreover, when the chimera was used as a seed parent, most of the seedlings were tetraploids.

**Table 4.** Comparison of morphological characteristics of flowers and pollen grains in the diploid, the tetraploid, the Yubeni, and the chimera of the Meiwa kumquat.

| | Ploidy Level | Flower Bud (mm) | | No. of Petal | Petal (mm) | | Length of Pistil (mm) | Ovary (mm) | | No. of Stamens | Pollen Grain (μm) | | Shape Index of Pollen Grain [y] | Pollen Stability Rate (%) | Pollen Germination Rate (%) |
| | | Length | Width | | Length | Width | | Diameter | Height | | Length | Width | | | |
|---|---|---|---|---|---|---|---|---|---|---|---|---|---|---|---|
| Diploid | 2x | 8.8 c[z] | 5.9 b | 5.2 | 8.9 b | 4.1 b | 5.3 b | 2.0 b | 2.1 c | 16.3 | 29.5 c | 17.4 c | 1.7 a | 97.5 a | 37.2 a |
| Tetraploid | 4x | 10.1 b | 7.5 a | 5.1 | 9.7 b | 4.8 a | 5.5 b | 2.4 a | 2.5 b | 17.9 | 38.1 a | 29.1 a | 1.3 b | 85.8 b | 18.0 b |
| Yubeni | 2x+4x | 11.7 a | 7.5 a | 5.1 | 11.0 a | 5.3 a | 6.1 a | 2.5 a | 3.0 a | 18.5 | 36.6 ab | 25.2 b | 1.4 b | 72.4 c | 12.5 b |
| Chimera | 2x+4x | 10.9 ab | 7.1 a | 4.9 | 10.8 a | 5.1 a | 5.6 b | 2.4 a | 2.5 b | 16.9 | 35.2 b | 25.4 b | 1.4 b | 80.2 bc | 14.2 b |

[z] Different letters represent significant differences in Tukey's multiple test, 1% level. [y] Length of pollen grain/Width of pollen grain.

**Table 5.** Comparison of morphological characteristics of fruits in the diploid, the tetraploid, the Yubeni, and the chimera of the Meiwa kumquat.

| | Ploidy Level | Fruit Wt. (g) | Fruit (mm) | | Shape Index of Fruit [z] | Pericarp Wt. (g) | Pericarp Wt./Fruit Wt. (%) | No. of Locules | No. of Developed Seeds/Fruit | No. of Undeveloped Seeds/Fruit | Developed Seeds/Fruit (%) | SSC (°Brix) | | TA of Pericarp (%) |
| | | | Diameter | Height | | | | | | | | Pericarp | Juice Sac | |
|---|---|---|---|---|---|---|---|---|---|---|---|---|---|---|
| Diploid | 2x | 22.2 | 33.6 | 35.9 | 93.7 | 12.6 b [y] | 62.3 c | 5.7 | 5.5 a | 0.9 | 86.8 ab | 16.3 c | 14.0 b | 0.31 |
| Tetraploid | 4x | 22.7 | 33.7 | 35.1 | 96.1 | 15.9 a | 74.5 a | 6.4 | 3.8 b | 1.5 | 74.2 b | 19.8 ab | 17.8 a | 0.26 |
| Yubeni | 2x+4x | 24.8 | 35.0 | 36.2 | 96.3 | 18.3 a | 68.5 b | 6.1 | 2.1 c | 0 | 100 a | 19.5 b | 16.0 ab | 0.30 |
| Chimera | 2x+4x | 23.4 | 33.8 | 35.6 | 94.8 | 16.3 a | 70.4 b | 6.1 | 2.6 c | 0.3 | 98.3 a | 21.3 a | 17.8 a | 0.27 |

[z] (Diameter of fruit/height of fruit) × 100. [y] Different letters represent significant differences in Tukey's multiple test, 1% level. Wt.: weight.

**Table 6.** Seed content and ploidy level of the seedlings obtained from the reciprocal crosses between the diploid and the chimera of the Meiwa kumquat.

| Cross Combination | | No. of Fruits | No. of Seeds | | No. of Developed Seeds/Fruit | Developed Seeds (%) [y] | Av. Developed Seed Weight | No. of Seedlings Examined | Ploidy Level | | |
| Seed Parent | Pollen Parent | | Developed | Undeveloped | | | | | 2x | 3x | 4x |
|---|---|---|---|---|---|---|---|---|---|---|---|
| Diploid | Self-pollination | 10 | 55 | 9 | 5.5 | 86.8 | 122.0 | 50 | 50 | 0 | 0 |
| | Chimera | 20 | 27 | 90 | 1.4 * | 23.1 * | 97.5 * | 77 | 68 | 8 | 1 |
| Chimera | Self-pollination | 10 | 33 | 4 | 3.3 | 89.2 | 148.9 | 50 | 50 | 0 | 0 |
| | Diploid | 10 | 26 | 4 | 2.6 * | 86.7 NS [z] | 64.9 * | 72 | 0 | 12 | 60 |

[z] NS: no significant difference. * mean is significantly different at 1% levels by *t*-test. [y] (No. of developed seeds/No. of seeds) × 100. Av.: average.

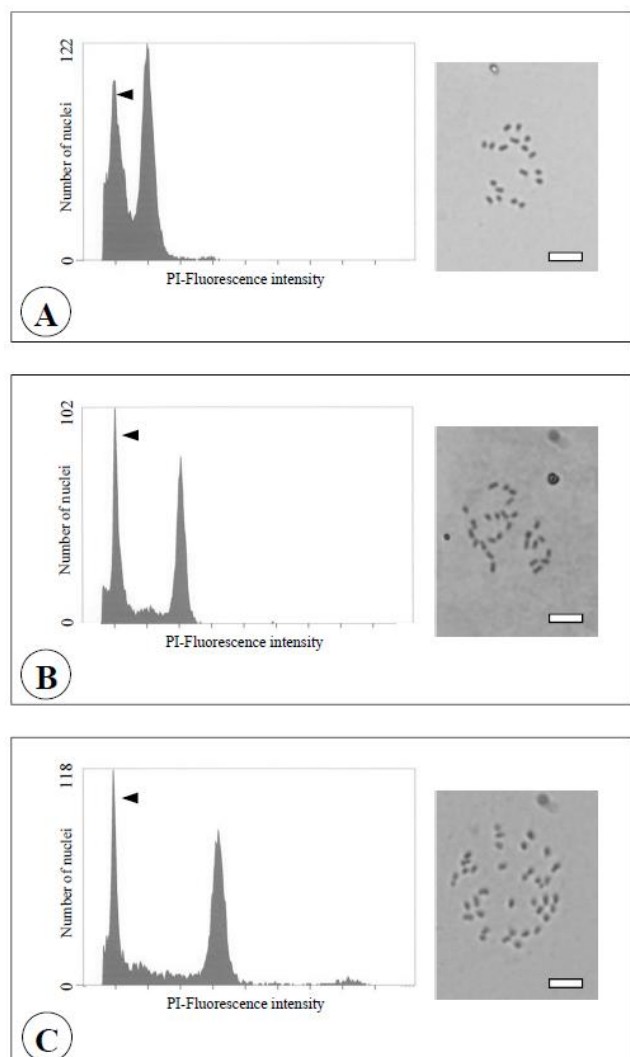

**Figure 7.** Flow cytometric analysis (left) and chromosome observation (right) of the seedlings obtained from reciprocal crosses between the diploid and the chimera of the Meiwa kumquat. Bars = 10 μm. **A**: Diploid (2n = 2x = 18). **B**: Triploid (2n = 3x = 27). **C**: Tetraploid (2n = 4x = 36). Arrows indicate the relative fluorescence of internal standard (the haploid pummelo (2n = x = 9, Yahata et al., [19])).

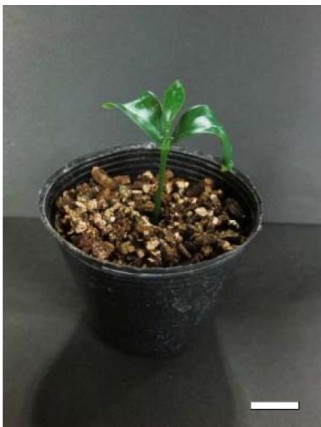

**Figure 8.** The triploid seedling obtained from the cross between the diploid and the chimera of the Meiwa kumquat (bar = 1 cm).

## 4. Discussion

In previous studies, the composition of each histogenic layer in periclinal chimeras such as the ploidy and the graft chimera was evaluated by morphological examination [11,12,20], histological observation [8–10], componential analysis [8], FCM analysis [12], and molecular biological techniques [7,9,10]. To clarify the ploidy level of each histogenic layer in the 2x+4x ploidy chimera, in the present study, flow cytometric analysis, cell observation, and morphological examination of several tissues and organs were carried out. Additionally, the ploidy chimera was crossed with the diploid kumquat to evaluate its reproductive organs of the ploidy chimera.

L1 is distinguished by the dermal system [3]. Especially, the size and the density of the guard cells were employed as the index to confirm the ploidy level in several plant species [19,20]. Epidermis cell size, guard cell size, and guard cell density of mature leaves in the chimera were mostly equal to those of the diploid control. This result showed that the chimera's ploidy level in L1 was diploid.

In the genus *Citrus* and its related genera, on the other hand, the juice sac is often used to analyze the origin of L1 [8,9,12]. Zhang et al. [8] demonstrated the origin of L1 by investigation of the carotenoid composition of the juice sac in periclinal graft chimera "Zaohang" navel orange composed of Robertson navel orange and Satsuma mandarin. Yasuda et al. [12] easily showed, by FCM analysis of the juice sac, that L1 ploidy level of the ploidy chimera Yubeni consists of diploid and tetraploid cells. In another study, however, Sugawara et al. [9] used RAPD analysis on a periclinal graft chimera composed of "Hamlin" sweet orange and Satsuma mandarin to show that cells derived from L1 and L2 were involved in the development of the juice sac. As reported by Nii and Coombe [21], the juice sac develops from epidermal cells and sub-dermal layers. In the present study, we could detect a single diploid peak just like a previous report by Yasuda et al. [12]. From here onwards, it has been considered that the juice sac in kumquats mainly differentiated from L1.

For L2 to differentiate into parenchymas and reproductive organs, mesophyll cells, pollen, and seeds are often used for the analysis of its origin [9,11,12]. The pollen of the chimera in the present study had as much size and fertility as the tetraploid and the Yubeni, and tetraploid and triploid progenies appeared in the cross between the chimera and diploid kumquat. Furthermore, the sizes of the subepidermal cells in the leaves of the chimera were significantly larger than those of the diploid, and were similar to those of the tetraploid and the Yubeni. For these reasons, it was considered that the L2 of the chimera was tetraploid.

Regarding L3, which is differentiated into cambium and pith, Yasuda et al. [12] presumed the ploidy level of L3 in the Yubeni by FCM analysis of the midrib. The FCM analysis of the midrib of the chimera showed tetraploidy. In the cell observations of the vessels and sieve tubes using histological techniques, furthermore, the cells of the chimera were the same size as the tetraploid and the Yubeni. Definitely, it was shown that the polyploidy of L3 in the chimera was tetraploid.

The chimera used in the present study was supposed to be a ploidy periclinal chimera, with diploids in the outermost layer (L1) and tetraploids in the inner layers (L2 and L3) of the shoot apical meristem. The morphological characteristics of the chimera were similar to that of the tetraploid Meiwa kumquat as previously reported [14,22,23]. Especially, it was reported that the fruit of these tetraploid kumquats had a thicker pericarp and a higher sugar content than the fruit of the diploid ones. The albedo, which is the main edible part of kumquats, differentiates from L2 [3], so Yasuda et al. [12] showed that the tetraploidization of L2 added to the superior fruit quality of the ploidy periclinal chimera Yubeni (diploids in L1 and tetraploids in L2 and L3). Because the chimera in the present study also had desirable fruit traits for kumquats, such as a thick pericarp, a high sugar content, and a small number of developed seeds, ploidy periclinal chimeras with tetraploids in L2 are very useful for kumquats. Furthermore, in the present study, triploid progenies were obtained from reciprocal crosses with the diploid kumquat. This result indicates that ploidy periclinal chimeras with tetraploids in L2 can be useful as parents for triploid breeding, where seedless fruits can be expected.

In conclusion, the 2x+4x ploidy chimera was confirmed as a ploidy periclinal chimera with diploids in L1 and tetraploids in L2 and L3. We plan to carry out the research not only on the

utilization of the 2x+4x ploidy chimera for triploid breeding but also on the commercial growing of its chimera. Additionally, we need to develop an efficient production technique for the 2x–4x–4x ploidy chimera type.

**Author Contributions:** M.Y., A.T., H.M., K.Y., and H.K. conceived and supervised research. T.N., M.S., and M.Y. designed experiments. T.N. conducted most experiments. M.S. and T.O. assisted and performed some experiments. M.S. analyzed data and wrote the manuscript. M.Y., A.T., H.M., K.Y., and H.K. supervised the preparation of the manuscript.

**Conflicts of Interest:** The authors declare no conflict of interest.

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
