# Peer review of "The Confirmation of a Ploidy Periclinal Chimera of the Meiwa Kumquat (Fortunella crassifolia Swingle) Induced by Colchicine Treatment to Nucellar Embryos and Its Morphological Characteristics"

_agronomy, doi:10.3390/agronomy9090562_

Round 1
Reviewer 1 Report
Abstract should include some reference to usefulness of the ploidy periclinal chimera in future breeding
Line 52.. delete 'was'.....cultivars were (remove 'was')registered in Japan
L61 kunquat could be used "both" for the parental line Substitute 'both' in place of 'not only'
L63,64 rephrase... to prove future usefulness.... remove "in the future"
Figure 1. Include Meiwa Kumquat in front of the word chimera in the title
Table 5. What is the result of statistical analysis of Fruit wt., fruit size and Shape index? Include in table.
L271 change to chimera was 'confirmed' as a ploidy perclinal chimera
L270 Expand on what is the usefulness of obtaining triploid progeny and significance in breeding
Author Response
Thank for revising the paper and giving me so many constructive advices from reviewers. We acknowledge very much the critical comments given by the reviewers. We accepted these comments and revised our maniscript.
Abstract should include some reference to usefulness of the ploidy periclinal chimera in future breeding
→We accepted these comments and added the sentence. (L32-35)
Line 52.. delete 'was'.....cultivars were (remove 'was')registered in Japan
→We revised this part. (L56)
L61 kunquat could be used "both" for the parental line Substitute 'both' in place of 'not only'
→We revised this part. (L65-66)
L63,64 rephrase... to prove future usefulness.... remove "in the future"
→We revised this part. (L67)
Figure 1. Include Meiwa Kumquat in front of the word chimera in the title
→We revised this part. (L142)
Table 5. What is the result of statistical analysis of Fruit wt., fruit size and Shape index? Include in table.
→We did not list because there was no significant difference in Tukey's multiple test. As you indicated, we added NS (no significant difference). (L199-201)
L270 Expand on what is the usefulness of obtaining triploid progeny and significance in breeding
→We accepted these comments and modified the sentence. (L273)
L271 change to chimera was 'confirmed' as a ploidy perclinal chimera
→We revised this part. (L274)
I am looking forward your response.
Yours sincerely
Reviewer 2 Report
Overall, this is well-written manuscript. The introduction is relevant. The approach to experimentation and methodology used seem appropriate. The results are clearly presented and well discussed. The manuscript could be accepted after minor revision. Specific comments follow:
Introduction Page 2 line 44 For better clarity explain three types of chimeras.
Materials and Methods
2.1. Plant materials. The origin of ploidy chimera is not clear, please add references or information on the source of the plant.
Page 3 line 99 Please add reference or describe the preparation of the material for scanning electron microscope.
Page 3 line 101 Please add the name of the program which were used for cell measurements.
Results
Page 4 Line 150 and Page 5 Fig. 2B. There is an information in the text that the chimera had significantly larger flower buds and ovaries as compared to those of the diploid. It is not so clear when compared with Fig. 2B. I can see the difference in flower shape rather than size.
Author Response
Thank for revising the paper and giving me so many constructive advices from reviewers. We acknowledge very much the critical comments given by the reviewers. We accepted these comments and revised our maniscript.
Introduction
Page 2 line 44 For better clarity explain three types of chimeras.
→We accepted these comments and added the sentence. (L46-48)
Materials and Methods
2.1. Plant materials. The origin of ploidy chimera is not clear, please add references or information on the source of the plant.
→We added the reference. (L83)
Page 3 line 99 Please add reference or describe the preparation of the material for scanning electron microscope.
→Since this SEM model does not require pretreatment of samples (fixation, dehydration, evaporation coating…), the raw samples can be observed directly. In most experiments using this SEM model, details are not described in the papers.
Page 3 line 101 Please add the name of the program which were used for cell measurements.
→We accepted these comments and added the sentence. (L108)
Results
Page 4 Line 150 and Page 5 Fig. 2B. There is an information in the text that the chimera had significantly larger flower buds and ovaries as compared to those of the diploid. It is not so clear when compared with Fig. 2B. I can see the difference in flower shape rather than size.
→As you pointed out, the difference in size is difficult to understand. But, we think there is a clear difference in the size of the flower buds. We looked for an alternative photo but found no better picture than this photo. I would like this photo in this paper. I'm sorry.
I am looking forward your response.
Yours sincerely